# A Quantitative Chemical Method for Determining the Surface Concentration of Stone–Wales Defects for 1D and 2D Carbon Nanomaterials

**DOI:** 10.3390/nano12050883

**Published:** 2022-03-07

**Authors:** Alexander Voznyakovskii, Anna Neverovskaya, Aleksei Vozniakovskii, Sergey Kidalov

**Affiliations:** 1Institute for Synthetic Rubber, 198035 Saint Petersburg, Russia; voznap@mail.ru (A.V.); anna-neverovskaya@yandex.ru (A.N.); 2Ioffe Institute, 194021 Saint Petersburg, Russia; alexey_inform@mail.ru

**Keywords:** few-layer graphene, structural defects, self-propagating high-temperature synthesis, Stone–Wales defects, graphene nanostructures, carbon nanotubes, reduced graphene oxide

## Abstract

A quantitative method is proposed to determine Stone–Wales defects for 1D and 2D carbon nanostructures. The technique is based on the diene synthesis reaction (Diels–Alder reaction). The proposed method was used to determine Stone–Wales defects in the few-layer graphene (FLG) nanostructures synthesized by the self-propagating high-temperature synthesis (SHS) process in reduced graphene oxide (rGO) synthesized based on the method of Hammers and in the single-walled carbon nanotubes (SWCNT) TUBAL trademark, Russia. Our research has shown that the structure of FLG is free of Stone–Wales defects, while the surface concentration of Stone–Wales defects in TUBAL carbon nanotubes is 1.1 × 10^−5^ mol/m^2^ and 3.6 × 10^−5^ mol/m^2^ for rGO.

## 1. Introduction

The awarding of the Nobel Prize to A. Geim and K. Novoselov for their work on exfoliation and prediction of the properties of 2D nanocarbon-graphene for many research groups was the impetus for the start of research on the fine organization of the nanocarbon family with sp^2^ hybridization of carbon atoms (the family of graphene structures, including, of course, graphene itself-2D nanocarbons, single-walled and multi-walled nanotubes-1D nanocarbons). First of all, the surge of this interest is because atomic-scale defects (impurities, vacancies, topological defects) significantly affect nanocarbon particles’ physical and electrical parameters [1]. As a rule, most researchers focus on the detection of Stone–Wales (SW) defects, which make the greatest contribution to the deviation of the electrophysical properties of defective 1D and 2D nanocarbons from those of nanocarbons with an undamaged structure [2]. The SW defect is formed when one of the C–C bonds in the monolayer plane is rotated by an angle of 90° (Stone–Wales transformation), which leads to the appearance of two heptagons and two pentagons. The increased interest in the study of SW defects is also because, due to the low formation energy shown (5 eV), the formation of SW defects are energetically more favorable and, therefore, the probability of their formation is higher than the probability of formation of other topological defects (reference [3] and references in it). It should be noted that SW defects are not a deterministic value, that is, it depends both on the nature of the starting material (for example, the imperfection of graphite) and on the specifics of the production method (for example, various versions of the Hummers method) when obtaining 2D graphene structures. At the same time, since SW defects are atomic-scale defects, their determination and characterization comprise an extraordinary task and require the use of complex experimental techniques complex and expensive instrumentation [4,5,6,7]. This is due to the fact that the known methods for determining SW defects are qualitative or allow only comparative results to be obtained.

To the best of our knowledge, there is no description of the methods in the literature that would allow one to obtain quantitative data on the content of SW 1D/2D in nanocarbons. Nevertheless, the possibility of getting such data will be very useful in classifying the families of 1D and 2D nanocarbons, taking into account quantitative data on the content of SW defects, and choosing promising areas for their application.

This work aimed to develop a publicly available and efficient method for the quantitative determination of SW defects content in 1D and 2D nanocarbons.

## 2. Investigated Materials

For the study, we used FLG synthesized from starch under the conditions of the SHS process (FLG-SHS) [8], single-walled carbon nanotubes (SWCNT, TUBAL trademark, JSC “OCSiAl”, Novosibirsk, Russia) (https://tuball.com/additives (accessed on 1 March 2022)) and GO synthesized by a modified Hammers method [9], which was treated with hydrazine to obtain reduced graphene oxide (rGO).

## 3. Methods Used for Analysis and Experimental Technique

The images were obtained by scanning electron microscopy on a TESCAN Mira-3M, SEM Supra55VP-3259 microscopes (Brno, Czech) and transmission electron microscopy on a 50 kV FEI Tecnai G2 30 S-TWIN microscope (Hillsboro, OR, USA).

In the TEM study, the powder samples were placed in ethanol, sonicated for 5 min, and mounted on a carbon grid.

The quality of synthesized samples was estimated using Raman spectra recorded on a Confotec nr500 (532 nm, SOL Instruments, Minsk, Belarus).

Specific surface areas of synthesized samples were determined using multilayer adsorption on an ASAP 2020 analyzer (Norcross, GA, USA). Nitrogen was used as the adsorbate. The sample preparation was performed according to the standard procedure of heating the samples in a vacuum at 300 °C for 3 h before the measurements. The measurement error did not exceed 3%.

Chromatographic studies were carried out using a Clarus 500 gas chromatograph. Research parameters: column temperature −145 °C; detector temperature −250 °C; evaporator temperature −250 °C; gas rate −30 mL/min.

## 4. Stone–Wales Defects

Stone–Wales defect is a crystallographic defect in carbon nanotubes, graphene, and other crystals with a hexagonal crystal lattice appearing when one of the C–C bonds is rotated through an angle of 90°, as a result of which four hexagons of carbon atoms are converted into two heptagons and two pentagons [10]. Because of this rearrangement, active dienophilic vacancies are formed in the structure of nanotubes and graphene. In the practice of organic chemistry, active dienophilic vacancies are used to obtain cyclic compounds by the reaction of the so-called “diene synthesis”—the reaction of [4 + 2]-cycloaddition (Diels–Alder reaction) [11]. The Diels–Alder reaction is a coordinated [4 + 2] cycloaddition occurring between a 1,3-diene and an unsaturated compound, a dienophile. Usually, a diene contains an electron-donor substituent, and a dienophile an electron-withdrawing group. It is known that the walls and ends of a carbon nanotube contain defective elements (five-membered cycles) that can act as dienophiles [12]. The presence of such defects in single-walled carbon nanotubes (SWCNTs) was shown using the reaction with α-methylstyrene. The last one was selected as a conjugated diene due to the fact that, unlike classical dienes—cyclopentadiene and styrene—it does not form homopolymer. Diene synthesis reaction scheme is presented by the example of the reaction of alpha α-methylstyrene with the surface of single-walled nanotubes in Figure 1.

According to the diene synthesis scheme, the formation of cyclic compounds is a thermodynamically favourable reaction; therefore, the reaction proceeds irreversibly and, accordingly, quantitatively. This nature of the reaction makes it possible to use it for the quantitative determination of possible Stone–Wales defects in carbons nanostructures, the surface of which is formed by carbon atoms with sp^2^ hybridization.

The progress of the reaction was monitored by the method of gas–liquid chromatography (GLC). To carry out the experiment, a mixture of α-methylstyrene (main reagent) with o-xylene (standard) was added to a suspension of SWCNTs in toluene with vigorous stirring. The resulting suspension was placed on a magnetic stirrer. Samples of the mixture taken every 4 h were introduced into the chromatograph and the ratio of α-methylstyrene/o-xylene in the mixture was determined. From the ratio of the areas of the o-xylene/α-methylstyrene peaks for each sample, it was concluded that the reaction was progressing. The criterion for the progress of the reaction was a sequential decrease in the content of α-methylstyrene in the suspension. The calculated degree of addition of α-methylstyrene to the SWCNT surface obtained by us was 28.2 wt %.

To eliminate the systematic error of the experiment, we specially set up a blank experiment, which showed the absence of sorption of o-xylene on the surface of the selected series of nanocarbons.

## 5. Results and Discussion

Figure 2 shows electronic images of SWCNTs obtained by SEM and TEM methods.

As shown in Figure 2, SWCNTs are tangled aggregates in the form of tangles consisting of individual CNTs. Such a material structure is typical for all CNTs synthesized in the form of a powder by the CCVD method. The average length of SWCNTs exceeds 200 nm, but the diameter of individual SWCNTs does not exceed 10 nm. However, as can be seen in Figure 2c, in addition to single-walled CNTs, the sample contains a certain amount of few-walled CNTs.

Figure 3 shows electron images of FLG-SHS obtained by SEM and TEM methods.

As shown in Figure 3a, the FLG-SHS have planar dimensions up to several tens of microns. As shown in Figure 3b, the synthesized particles have a few-layer structure formed by the superposition of differently oriented graphene layers. It is also seen that the number of layers in the samples does not exceed five.

Figure 4 shows the Raman spectra of SWCNTs, FLG-SHS, and rGO.

The Raman spectrum of SWCNTs contains D peak (1345 cm^−1^), G peak (1590 cm^−1^), and G‘peak (2680 cm^−1^), typical of CNTs. Similar spectra were observed in [13]. The intensity ratio of the D peak (I_d_) and the G peak (I_g_) is 0.028, which is usually attributed by researchers to the extremely low defectiveness of CNTs [14]. It should be noted that it is by the intensity of the D peak that researchers estimate the defectiveness of the sp^2^ structure of such materials as CNTs and graphene nanostructures. Using data on the position of the RBM peak, the average CNT diameter was estimated using the formula below [15].
(1)d=220/(υ−14)
where d–CNT diameter, nm; υ—RBM peak position, cm^−1^. 

It was found that the average diameter of CNTs should not exceed two nm. This calculation confirms that the fraction of few-walled CNTs in the sample is negligible and cannot seriously affect subsequent results.

Sample FLG-SHS exhibits D peak (1345 cm^−1^), G peak (1600 cm^−1^), and 2D peak (2500 cm^−1^) typical for graphene. Similar spectra were obtained in [16]. However, unlike CNTs, this material has a highly intense D peak, and the I_d_/I_g_ ratio is 1.2, which, as in the case of CNTs, is usually associated with a high defectiveness of the material. The Raman spectrum of rGO is similar to that of FLG. However, the I_d_/I_g_ ratio is 0.76, which suggests that this sample has less structural imperfection than FLG. Similar rGO spectra were also obtained in [17].

To check the data on the defectiveness of the structure of CNT, FLG, and rGO samples obtained by Raman spectroscopy, we experimented on the possibility of including these materials in the diene synthesis reaction. It consisted of an attempt to functionalize them with α-methylstyrene. A mixture of α-methylstyrene and o-xylene taken in equal amounts was added to a suspension of carbon nanostructures in toluene to carry out the diene synthesis reaction.

Control over the passage of the reaction was carried out by gas–liquid chromatog-raphy (GLC). A mixture of α-methylstyrene (basic reagent) with o-xylene (standard) was added to a suspension of SWCNT/rGO/FLG in toluene with vigorous stirring. The resulting suspension was placed on a magnetic stirrer. Samples of the mixture, taken every 3 h, were injected into the chromatography column, and the ratios of α-methylstyrene/o-xylene in the mixture were determined. According to the ratio of peak areas of o-xylene/α-methylstyrene for each sample, it was concluded that the reaction was proceeding. The criterion for the reaction was a consistent decrease in the content of α-methylstyrene in the suspension.

To eliminate the systematic error of the experiment, we specially set up a blank experiment (there was no α-methylstyrene in the solution), which showed the absence of sorption of o-xylene on the surface of the selected series of nanocarbons.

As shown by our carefully performed experiments, it can be argued that the diene synthesis reaction for FLG does not work. At a minimum, the impossibility of the diene synthesis reaction indicates the absence of defects of the Stone–Wales type or their existence at concentrations below the sensitivity of the registration method (gas–liquid chromatography).

Accordingly, the defectiveness of the structure of the FLG particles obtained by us, demonstrated by the nature of the Raman spectrum curve, can be associated exclusively with concentrated vacancy defects. It was of undoubted interest to carry out similar experiments to determine the Stone–Wales defects for SWCNT “TUBAL”.

The presence of defects in SWCNT “TUBAL” is well known, and therefore, in addition to purely practical interest, such work was necessary to verify the effectiveness of the proposed method independently. These experiments were carried out under conditions similar to the functionalization of FLG.

In contrast to FLG-SHS, the diene synthesis reaction was efficient for SWCNT and rGO. The concentration of Stone–Wales defects calculated by us for SWCNT and rGO turned out to be equal to C_sw_ = 3.3 × 10^−3^ mol/g and C_sw_ = 20.9 × 10^−3^ mol/g, respectively. Taking into account the specific surface area of nanomaterials (300 m^2^/g for SWCNT and 580 m^2^/g for rGO), the surface concentration of Stone–Wales defects is C_sw_ = 1.1 × 10^−5^ mol/m^2^ and C_sw_ = 3.6 × 10^−5^ mol/m^2^, respectively. 

The obtained value corresponds to the number of moles of α-methylstyrene irreversibly reacted with the surface of carbon nanostructures by the reaction of diene synthesis, which quantitatively corresponds to the concentration of dienophilic vacancies—Stone– Wales defects.

Summary information with data for various carbon nanomaterials studied by us is given in Table 1. The specific surface of carbon nanomaterials needed to calculate the surface SW defects concentration was calculated from BET data.

The obtained value of SW defects concentration corresponds to the number of moles of α-methylstyrene irreversibly reacted with the surface of carbon nanostructures by the reaction of diene synthesis, which quantitatively corresponds to the concentration of dienophilic vacancies—SW defects.

As regards the difference in defects of rGO and FLG, we attribute both to different mechanisms of their production (up-bottom/bottom-up) and different sources of their production. As a result, we obtained for the first time data on the quantitative value of SW defects in 1D and 2D carbon nanostructures, available for the first time.

## 6. Conclusions

A quantitative method based on the diene synthesis reaction was proposed to determine Stone–Wales defects in carbon nanostructures with sp^2^ hybridization of carbon atoms (graphene nanostructures, nanotubes). It is shown that from biopolymers under the conditions of the SHS process, it is possible to obtain FLG nanosheets that are free of Stone–Wales defects.

## Figures and Tables

**Figure 1 nanomaterials-12-00883-f001:**
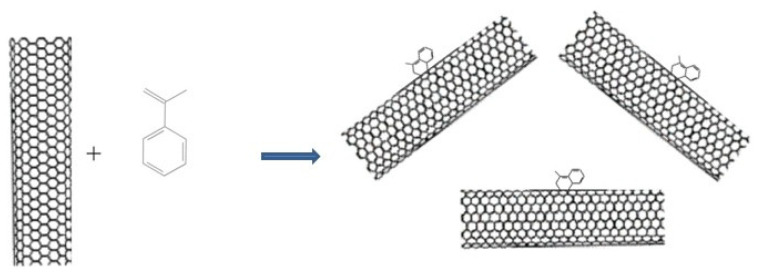
Scheme of joining α-methylstyrene to nanotubes.

**Figure 2 nanomaterials-12-00883-f002:**
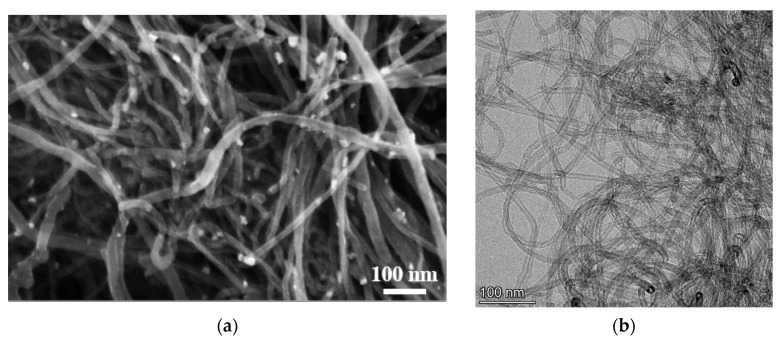
SEM (**a**) and TEM (**b**,**c**) images of CNT.

**Figure 3 nanomaterials-12-00883-f003:**
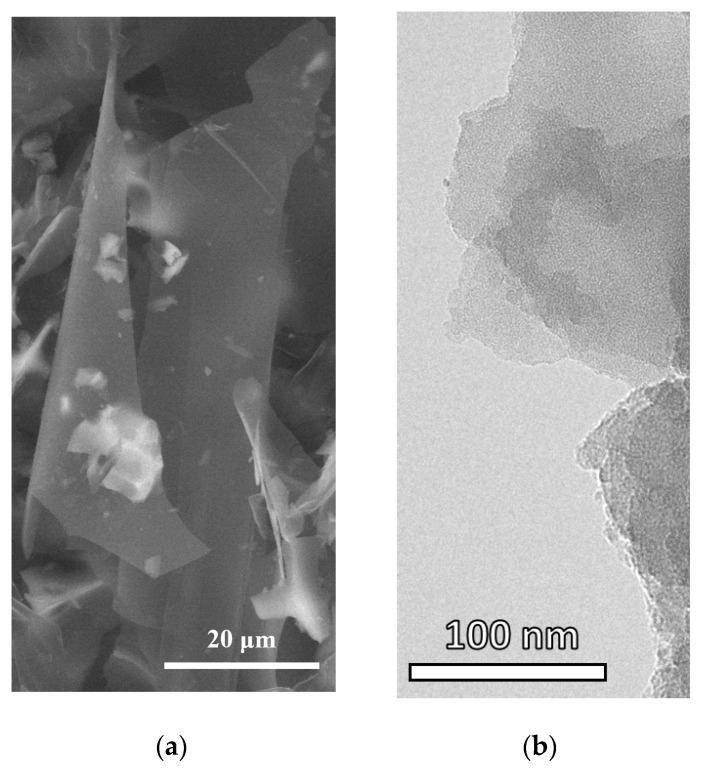
SEM (**a**) and TEM (**b**) images of FLG-SHS.

**Figure 4 nanomaterials-12-00883-f004:**
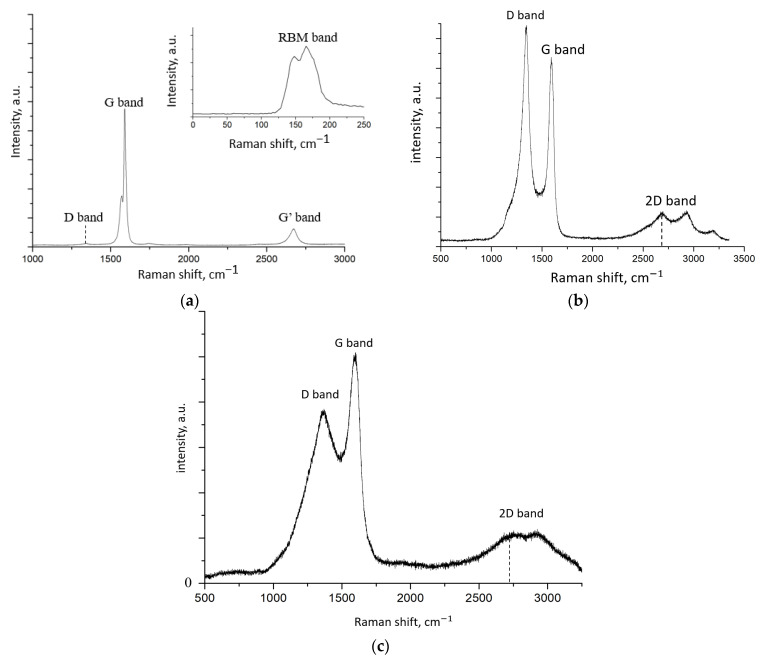
Raman spectra of SWCNT (**a**), graphene nanosheets, synthesized from starch under the conditions of the SHS process (**b**), rGO (**c**).

**Table 1 nanomaterials-12-00883-t001:** Parameters of studied carbon nanomaterials.

Sample	Stone–Wales Defects Concentration C_SW_ (mol/m^2^)	I_d_/I_g_	Specific Surface m^2^/g
SWCNT	1.1 × 10^−5^	0.028	300
rGO	3.6 × 10^−5^	0.76	580
FLG	0	1.2	660

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
