# Peer review of "A Quantitative Chemical Method for Determining the Surface Concentration of Stone–Wales Defects for 1D and 2D Carbon Nanomaterials"

_nanomaterials, 2022, doi:10.3390/nano12050883_

Round 1

Reviewer 1 Report

In this study, the authors proposed a quantitative method of determining Stone-Wales defects of different carbon nanostructures through the diene synthesis reaction. The authors claimed that the structure of FLG was free of Stone-Wales defects, while the surface concentration of Stone-Wales defects in TUBAL CNTs and rGO was 1.1×10-5 mol/m2 and 3.6×10-5 mol/m2, respectively. The reviewer would recommend this manuscript to be accepted for publication if the following concerns can be well addressed by the authors.

  1. As different nanocarbon materials including SWCNTs, FLG, rGO were investigated, the title of this paper should be revised.
  2. The correctness of SW defects measurement should be validated by other experimental or simulation methods.
  3. 2b showed the TEM image of CNTs adopted in this study. It is not SWCNT. Please check.
  4. The introduction section should be reorganized. The introduction of SHS method is suggested to move before the objective of this study.

Reviewer 2 Report

The authors performed Diels-Alder reaction on different kinds of carbon-based nanostructures, and characterized the materials simply by SEM/TEM and Raman. I cannot recommend the publication of the manuscript due to the following reasons.

Diels-Alder reaction has been already reported for different kinds of carbon-based materials including carbon nanotubes and graphene. The authors will need to discuss the previous examples objectively in the introduction part and highlight the novelty of their work.

The title of the manuscript did not fit to the main content of the experiment and should be corrected.

The current experimental data are too weak to support the claims form the authors and the discussions on the current data are too simple. It is suggested the authors also performed other experiments including IR, XPS, TGA.

The language writing can be further improved including the typos, and the style of the reference should be uniform.

Reviewer 3 Report

  1. The average diameter of SWNTs is around 5~10 nm. However, looking at Fig. 2b, it can be easily confirmed that the minimum diameter is 20 nm, and it is composed of multi-layer walls rather than single walls. TEM results alone show that this sample is not SWNTs, but MWNTs or thin-MWNTs. Authors need to more clearly define information about their samples. And if possible, Raman measurements have been required.
  2. The raman X-axis range needs to be expanded to identify the RBM of SWNTs in Figure 4a. You also need an explanation for this.
  3. A typo in the text needs to be corrected. (ex. table 1. rGO(?), rOG(?)) and Unit correction is required. (ex: Id/Ig =1,2 ->1.2)  "," is wrong.
  4. The author does not match the contents of the introduction with the results of the main text. The introduction needs to be revised to fit the purpose of the study.

Round 2

Reviewer 1 Report

All the comments have been well addressed by the authors. The manuscript is now acceptable for publication.

Author Response

Dear Reviewer,

Thanks for your comments and remarks.
This made the manuscript better.

Reviewer 3 Report

I am pleased to inform you that your MS has been accepted for publication.

Author Response

(The authors gave the same response as above.)
